

# Data-specific substitution models improve protein-based phylogenetics

João M. Brazão[1], Peter G. Foster[2] and Cymon J. Cox[1]

[1] Centro de Ciências do Mar, Universidade do Algarve, Faro, Algarve, Portugal
[2] Department of Life Sciences, Natural History Museum, London, United Kingdom

## ABSTRACT

Calculating amino-acid substitution models that are specific for individual protein data sets is often difficult due to the computational burden of estimating large numbers of rate parameters. In this study, we tested the computational efficiency and accuracy of five methods used to estimate substitution models, namely Codeml, FastMG, IQ-TREE, P4 (maximum likelihood), and P4 (Bayesian inference). Data-specific substitution models were estimated from simulated alignments (with different lengths) that were generated from a known simulation model and simulation tree. Each of the resulting data-specific substitution models was used to calculate the maximum likelihood score of the simulation tree and simulated data that was used to calculate the model, and compared with the maximum likelihood scores of the known simulation model and simulation tree on the same simulated data. Additionally, the commonly-used empirical models, cpREV and WAG, were assessed similarly. Data-specific models performed better than the empirical models, which under-fitted the simulated alignments, had the highest difference to the simulation model maximum-likelihood score, clustered further from the simulation model in principal component analysis ordination, and inferred less accurate trees. Data-specific models and the simulation model shared statistically indistinguishable maximum-likelihood scores, indicating that the five methods were reasonably accurate at estimating substitution models by this measure. Nevertheless, tree statistics showed differences between optimal maximum likelihood trees. Unlike other model estimating methods, trees inferred using data-specific models generated with IQ-TREE and P4 (maximum likelihood) were not significantly different from the trees derived from the simulation model in each analysis, indicating that these two methods alone were the most accurate at estimating data-specific models.
To show the benefits of using data-specific protein models several published data sets were reanalysed using IQ-TREE-estimated models. These newly estimated models were a better fit to the data than the empirical models that were used by the original authors, often inferred longer trees, and resulted in different tree topologies in more than half of the re-analysed data sets. The results of this study show that software availability and high computation burden are not limitations to generating better-fitting data-specific amino-acid substitution models for phylogenetic analyses.

Corresponding author
Cymon J. Cox, cymon@ualg.pt

## INTRODUCTION

Maximum likelihood (ML) and Bayesian inference (BI) phylogenetic methods include a set of assumptions about the evolutionary process of change in molecular sequences (amino acids, nucleotides, or codons) that are specified by a substitution rate model. The resulting phylogenetic trees (topology and branch lengths) are dependent on the model, and a poor fit of the model to the data will affect the accuracy of tree reconstruction (*Keane et al., 2006*; *Cox & Foster, 2013*). Substitution rates at sites are assumed to follow a continuous-time Markov chain model: sites evolve independently of each other through time and are described by a probability of change at any particular site, where the states of the chain, and the probability of change from the current state, do not depend on the past states (*Felsenstein, 2004*; *Yang, 2014*). For analyses of proteins, an amino-acid substitution model is usually expressed as a $20 \times 20$ instantaneous rate matrix, where each off-diagonal element is the product of the relative rate of exchange between amino acids and the equilibrium frequency of the resulting amino-acid (*Swofford et al., 1996*). Typically, only 189 instantaneous rate parameters are considered, because the evolutionary process is assumed to be reversible at the same rate (a time-reversible process). Change in the substitution rate among sites is typically accommodated by modeling the distribution of rates with a discrete gamma-distribution (*Yang, 1994*).

During phylogenetic tree reconstruction it is expected that the substitution rates specified in the model are a good fit to the evolutionary process underlying change in the sequence data being analysed. Traditionally, large sets of proteins were used to calculate general models which were then used to analyse new data—these general-fitting, empirical, models were generated for analyses of particular genomes, or taxon groups. The use of empirical models is especially important when the new data to be analysed are of limited size and therefore unlikely to be sufficient to estimate all the substitution model rate parameters. Moreover, in the past, analyses of large data sets appropriate for calculating data-specific substitution rates, imposed a considerable computational burden. Nevertheless, despite today there being a general increase in the size of protein data sets used in phylogenetics, and the availability of faster computers with more efficient algorithms for calculating substitution models, the use of pre-computed, empirical models for the analysis of amino-acid sequences is still almost ubiquitous in phylogenetic practice.

The first widely-used amino-acid substitution models, namely the point accepted mutation (PAM) matrices (*Dayhoff, Schwartz & Orcutt, 1978*) and the JTT model (*Jones & Thornton, 1992*), were derived from nuclear protein data using parsimony counting methods. By contrast, the mitochondrial mtREV model (*Adachi & Hasegawa, 1996*) and the chloroplast cpREV model (*Adachi et al., 2000*), were estimated using ML but on smaller data sets (<100,000 amino-acids) due the computational burden of optimising both substitution rates and branch lengths simultaneously. The nuclear genomic models, WAG (*Whelan & Goldman, 2001*) and LG (*Le & Gascuel, 2008*), were also estimated using ML but with far larger data sets (~900,000 and ~6.5 million amino-acid residues, respectively), although optimisation procedures were simplified again to reduce computational burden. The only known amino-acid substitution model to have estimated

using Bayesian Markov chain Monte Carlo (MCMC) routines is the gcpREV calculated for chloroplast data of Streptophyta plants (*Cox & Foster, 2013*). The gcpREV model was shown to have a better fit to Streptophyta plant data than the more general plant cpREV model which was calculated with the inclusion of red algae. Likewise, protein-specific substitution models were found to be a better fit to protein virus data than the available empirical substitution models (*Del Amparo & Arenas, 2022*). These analyses demonstrate a common expectation that a model calculated specifically for the data-at-hand is going to be a better fit to the data than a more general-fitting model which might have wider application but less specificity, and therefore result in a theoretically better justified phylogenetic hypothesis.

With the ever-increasing availability of sequence data due to improvements in sequencing technologies, allied with the increased computational performance of modern computers, more than ever before phylogeneticists have the possibility of calculating and using data-specific protein substitution models instead of choosing from pre-computed empirical models. However, to date, the accuracy of existing methods for calculating amino-acid substitution models has not been evaluated collectively. In this study, we assess five methods for their ability to estimate accurate amino-acid substitution models: the methods are implemented in the programmes Codeml (PAML; *Yang, 1997*, *2007*), FastMG (*Dang et al., 2014*), IQ-TREE (*Nguyen et al., 2015*), and P4 (*Foster, 2004*). Each method estimates the 189 free rate parameters of a general time-reversible model using ML optimisation procedures. P4 can also estimate a substitution model using Bayesian methodology by sampling parameters from the posterior distribution of a MCMC.

To test the efficiency of the five methods, amino acid sequence data were simulated using a known amino-acid model (gcpREV) and a specified phylogenetic tree. Thereafter, the methods were tested with respect to their ability to calculate an amino-acid substitution model similar to the original simulation model. The new data-specific models were compared to the simulation model with respect to the likelihood scores of the simulation tree, clustering distance (principal component analysis (PCA)), and the similarity of the topology and branch lengths of reconstructed optimal trees with respect to the simulation tree. The analyses using data-specific models were also compared to those using the commonly-used empirical models (cpREV and WAG). Additionally, data-specific amino-acid models were estimated for a set of published phylogenetic analyses and the results using the new models compared with the results from the chosen models used in the original studies.

## MATERIALS AND METHODS

### Analyses of simulated sequence data using data-specific models and assessing five model estimation methods

Amino-acid sequence alignments were generated in P4 (vers. 1.3) by simulating sequence evolution using the gcpREV substitution model and a 26 taxon tree (taken from *de Sousa et al., 2019*) with fixed branch lengths (total tree length 7.74 expected numbers of substitutions per site). The simulation process consisted of generating a random root

sequence and evolving it over the simulation tree under the process specified by a simulation model (*Foster, 2004*). The gcpREV substitution model was used as simulation model. The root sequence had the model-specified gcpREV composition ($F_{mod}$) and sites were evolved under a discrete gamma-distribution of among-site rate variation, discretised with four categories ($\Gamma_4$). A total of 100 simulated alignments were generated with each of 400, 1,500, and 8,000 site lengths, which corresponds to a mean expected number of substitutions per branch of 63.2, 236.9, and 1,263.6 respectively. It should be noted that we do not consider the effect of incomplete sequences on model selection or reconstruction in this study.

The methodology used to assess the accuracy of the methods for calculating data-specific models is shown in Fig. S1. Data-specific amino-acid substitution models were estimated using ML optimisation in Codeml (PAML, vers. 4.9i), FastMG (vers. beta), IQ-TREE (vers. 1.6.8), and P4, and by BI in P4. Model parameters were calculated for a general time-reversible model (GTR; *Tavaré, 1986*), with $\Gamma_4$, and optimised composition frequencies ($F_{est}$), except when using Codeml where the composition frequencies are set to the empirical values of the data ($F_{emp}$). Model parameters were estimated using the simulation tree as a constraint (both topology and branch lengths) to reduce the variability in the methodology and enable a more direct comparison of calculating methods. Similarly, the method used in FastMG did not include the "alignment split algorithm" (use to reduce computation burden on calculating ML trees of large data sets) so that the estimation could be constrained to the simulation tree. For the Bayesian estimation of substitution model parameters MCMC analyses were run using P4 for 600,000 generations sampling parameter values every 100 generations, for a total of 6,000 samples. Of the posterior distribution samples, 1,000 were discarded as "burn-in", and the substitution rates calculated as the mean values of the remaining 5,000 samples. Exact commands and parameter values for each analysis are provided in the Data Availability Statement.

The data-specific models estimated using the five software methods are logical equivalents to the empirical models (WAG, gcpREV, and cpREV) to which they were compared in this study. In other words, all model comparisons had the same numbers of parameters (190 amino-acid fixed exchange rate parameters and 20 fixed amino-acid frequency parameters, and one free parameter for the alpha variable of $\Gamma_4$). The AIC (*Akaike, 1973*) or BIC (*Schwarz, 1978*) metrics are often used to compare substitution models when the numbers of parameters vary between models. However, because all the models compared in this study had the same number of parameters, the model-fit to the data was assessed using the log likelihood scores.

The estimated data-specific models and the commonly-used empirical models (cpREV and WAG) were visually compared to the simulation model (gcpREV) *via* ordination using PCA, after first normalizing the rate parameters of each data-specific model were normalized by dividing each value by the sum of the 189 parameters. Mean rate parameters were then calculated from each set of data-specific models calculated using each of the five methods and each of the three alignment lengths (400, 1,500, and 8,000 sites).

Data-specific models were compared to the simulation model with respect to their fit to the data. ML scores were calculated using IQ-TREE for each data-specific model (with $\Gamma_4$

and $F_{mod}$) on the same alignments the models were derived from, with the simulation tree as a constraint (topology and branch lengths). Similarly, ML scores using the equivalent models were calculated for gcpREV, cpREV, and WAG empirical models. The cpREV was chosen because it was determined by ModelFinder (implemented in IQ-TREE; *Kalyaanamoorthy et al., 2017*) as the best-fitting empirical model to the simulated alignments. By contrast, the WAG model, being derived from nuclear data, would be expected to have a lower fit to the data than cpREV. The statistically significant differences between ML scores were assessed using a two-tailed independent t-test (*Student, 1908*). Because the latter test assumes a normal distribution of the data, ML scores were tested using the Shapiro–Wilk test, which assessed the normality of the data (*Shapiro & Wilk, 1965*). The t-test null hypothesis assumed equal score means between the simulation model analyses and each data-specific model, cpREV, and WAG analyses. The null hypothesis was rejected with a $P$-value ($P$) significant at $<0.05$. However, a non-rejected null hypothesis using log-transformed data (where each variable x is replaced by log(x)), does not necessarily imply the same for the untransformed values, mainly when the variances of the log-transformed data are unequal (*Zhou, Gao & Hui, 1997*). Nevertheless, the analyses of log likelihood scores using the F-test of equality of variances (assess whether the variances between samples are equal) did not reject the assumption of equality of the variances between the gcpREV scores and each model scores, indicating that the analyses of the log likelihood values are likely in agreement with the untransformed likelihood values.

Tree reconstruction accuracy was tested for each data-specific model (with $\Gamma_4$ and $F_{mod}$) by comparing optimal ML tree topologies and branch lengths reconstructed with IQ-TREE with the simulation tree. The topological accuracy was assessed using the unweighted Robinson–Foulds (RF; *Robinson & Foulds, 1981*) and weighted Robinson–Foulds (WRF; *Robinson & Foulds, 1979*) tree metrics, as implemented in P4. The RF measures the number of branches that differ between the trees, while the WRF distance is the sum of the differences between all branch lengths. Additionally, the tree length (the sum of all branch lengths) of each optimised ML tree was compared to the simulation tree length. The statistically significant differences between tree distances were assessed using a two-tailed independent t-test, where the null hypothesis assumed a WRF mean distance and mean tree length difference equal to the simulation model tree results. The null hypothesis was rejected with a $p$ significant at $<0.05$. The assumption of normality of the data was assessed using the Shapiro–Wilk test. When the latter was rejected, the Wilcoxon test (*Wilcoxon, 1945*) was used instead of the t-test. The Wilcoxon test is a non-parametric test used to assess the null hypothesis of equality of the score means under the non-normality assumption.

### Re-analysis of published studies using data-specific models

To determine whether the use of data-specific amino-acid models would likely impact the results of phylogenetic analyses of empirical data, data-specific substitution models were estimated from published data sets using a GTR (with $\Gamma_4$ and $F_{est}$) model in IQ-TREE (Table S1). Optimal ML trees were inferred from the published data sets using the

data-specific models with the same among rate site variation parameters as used in the original studies. The resulting ML trees were compared to the original published trees with respect to their ML score, topological distance, and total tree length. The topological distances were calculated as a normalized RF (nRF) distance metrics (RF/RFmax, where RFmax is obtained by 2 * (number of taxa−3); *Kupczok, Haeseler & Klaere, 2008*). Where optimal ML trees and their scores were not available from the original publication, they were computed using the published trees and the original model.

## RESULTS

### Simulated sequence data analyses and assess five methods for estimating substitution models

Simulated sequence alignments were used to test the accuracy of five methods to generate data-specific amino-acid substitution models. The data-specific models were assessed by comparison to the simulation model with respect to ML scores calculated on the alignments the data-specific models were derived from, when using the simulation tree as a constraint (Table 1; Fig. S2). The ML scores from each set of analyses had a normal distribution according to the Shapiro–Wilk test and equal variance to the simulation model values. All five methods resulted in data-specific models that were a close fit to simulation model (gcpREV), that is, the mean ML scores of each set of alignment lengths were similar to the ML score of the simulation model. Besides, apart from those data-specific models of 8,000-site alignments estimated using FastMG, ML mean scores of data-specific models from all methods were higher than the simulation model mean scores. Data-specific models estimated using the P4-BI and FastMG method had the lowest difference to the simulation model score. The analyses using the P4-BI-estimated models had a mean score difference from the simulation model by 35, 55, and 82 log likelihood units, when inferred from 400, 1,500, and 8,000-site alignments respectively, while FastMG-estimated model analyses had a mean score difference by 73, 45, and −21 units. The P4-ML-estimated models had the highest mean scores and varying between 91 and 102 likelihood units when compared to gcpREV mean likelihood scores.

The differences of the Codeml- and IQ-TREE-estimated models to the simulation model were between 83 and 93 and between 81 and 88 likelihood units, respectively. Nevertheless, the likelihood scores resulting from data-specific model analyses were not significantly different from simulation model scores. By contrast, the mean ML scores derived from commonly-used empirical models were significantly lower than the simulation model scores ($p < 0.05$ as assessed by a two-tailed t-test; Table 1), apart of the cpREV analyses using the 400-site alignments. The mean likelihood scores calculated using the cpREV model deviated from the gcpREV analyses by −88, −332, and −1,770 units when inferred from 400, 1,500, and 8,000-site alignments respectively, while WAG mean scores deviated by −243, −903, and −4,837 likelihood units.

Accuracy of the estimated amino-acid models was also assessed by visually inspecting a PCA ordination between the gcpREV simulation model, cpREV and WAG empirical models, and the mean rates of the estimated data-specific models (Fig. 1). Mean-rate

**Table 1 Mean ML scores of each simulated sequence alignment (for three sets of alignment length) and its data-specific substitution rate model derived using five model estimation methods.**

| Amino-acid substitution models $(+\Gamma_4+F_{mod})$ | Mean ML scores of the simulation tree for 100 simulated data sets (log likelihood units) | | |
|---|---|---|---|
| | 400 sites ($\Delta$ simulation model) | 1,500 sites ($\Delta$ simulation model) | 8,000 sites ($\Delta$ simulation model) |
| gcpREV (simulation model) | −10,781 | −40,417 | −215,294 |
| cpREV | −10,869 (−88; 0.10) | −40,749* (−332; 0.0) | −217,064* (−1,770; 0.0) |
| WAG | −11,024* (−243; 0.0) | −41,318* (−902; 0.0) | −220,131* (−4,837; 0.0) |
| Codeml-estimated models | −10,698 (83; 0.12) | −40,326 (91; 0.37) | −215,201 (93; 0.63) |
| FastMG-estimated models | −10,708 (73; 0.17) | −40,372 (45; 0.66) | −215,315 (−21; 0.91) |
| IQ-TREE-estimated models | −10,700 (81; 0.13) | −40,329 (88; 0.38) | −215,209 (85; 0.66) |
| P4-ML-estimated models | −10,690 (91; 0.09) | −40,317 (100; 0.32) | −215,191 (102; 0.59) |
| P4-BI-estimated models | −10,745 (35; 0.52) | −40,361 (55; 0.58) | −215,212 (83; 0.67) |

**Note:**
ML scores were calculated using data-specific models $(+\Gamma_4+F_{mod})$ with the topology and branch lengths constrained to those of the simulation tree used to derive the simulated data. ML scores using the equivalent models were calculated for gcpREV, cpREV, and WAG empirical models for comparison. The null hypothesis of no difference between the mean ML scores of the simulation model (gcpREV) and the mean scores resulting from the estimated models, cpREV, and WAG was rejected with a $P$-value ($P$) significant at <0.05 (*), under a two-tailed t-test.

models were calculated from the mean of the normalised rates of data-specific models according to each alignment length. The cpREV and WAG models were two of the three most distant models from the simulation model (the other being the P4-BI-estimated 400 sites model), corroborating the likelihood score comparison analyses. By contrast, the mean models calculated from the Codeml- and P4-ML-estimated models clustered closest to the simulation model. The P4-BI-estimated mean rate models computed using the 1,500 and 8,000-site alignments were also relatively close to the simulation model, while the IQ-TREE derived models were more distant. FastMG mean models were clustered together further than the other mean rate models. Generally, the longer the simulated alignment, the closer the estimated model was to the simulation model in the ordination. However, this was not the case for Codeml- and FastMG-estimated models where the length of the simulated data had less effect on the overall accuracy of the estimated models.

The estimated data-specific models were also assessed with respect to their ability to reconstruct the topology and correctly estimate the branch of the simulation tree during an unconstrained tree search. The optimal ML trees inferred from 1,500 and 8,000 site simulated alignments always recovered the correct topology, *i.e.*, the same as the simulation tree. However, the trees inferred from the 400-site alignments using data-specific models failed in 19–22% of replicates to recover the correct topology, having a RF mean varying between 0.38 and 0.44 (Table S2). The cpREV and WAG derived trees
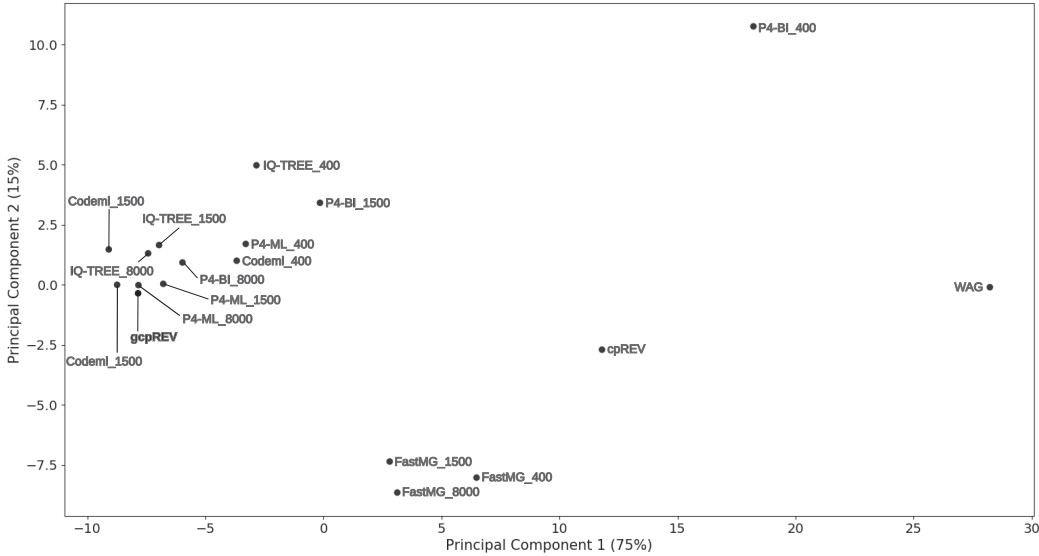

**Figure 1 Principal component analysis of model exchange values.** Ordination of the mean values of the normalised exchange rate model parameters of the data-specific models (named according to the calculation method and simulated data length), the gcpREV simulation model, and the cpREV and WAG empirical models.

topologically different to the simulation tree 24 (mean RF 0.48) and 25 times (mean RF 0.54) respectively, whereas the simulation model itself failed to recover the simulation tree 22 times (mean RF 0.44).

The WRF mean distances between the optimal ML trees and the simulation tree decreased as the length of the data set increased. The Shapiro–Wilk test indicated the normality of the data for each set of WRF distances, except for the 1500-site alignment analyses using the FastMG-estimated models. The cpREV and WAG derived trees had the highest WRF distances and were significantly different from the simulation (gcpREV) model derived statistics ($P < 0.05$; Fig. 2; Table 2). The WRF distances computed using the FastMG-estimated models had a higher difference to WRF means derived from the simulation model than the remaining data-specific models at all data lengths, and statistically higher when computed using the 1,500- and 8,000-site alignments. The P4-ML-estimated models had the lowest WRF mean distance within the analyses using the 1,500-site alignments. The analyses of the 8,000-site alignments using the IQ-TREE- and P4-estimated (ML and BI) models shared the lowest WRF mean distances overall and the closest to the simulation model. The WRF distances resulting from Codeml-estimated model analyses were the closest to the gcpREV values, when using the 400- and 1,500-site alignments. Nonetheless, the WRF mean distances computed using the optimal trees derived from the Codeml-, IQ-TREE-, and P4-estimated models were not significantly different to the simulation model results.

The optimal ML trees inferred using the cpREV and WAG models had the highest tree length differences to the simulation tree, did not converge with the simulation tree with longer alignments and were significantly shorter ($P < 0.05$ as assessed by a two-tailed t-test) than gcpREV derived trees (Table 2; Fig. 2). The mean length of the optimal trees inferred

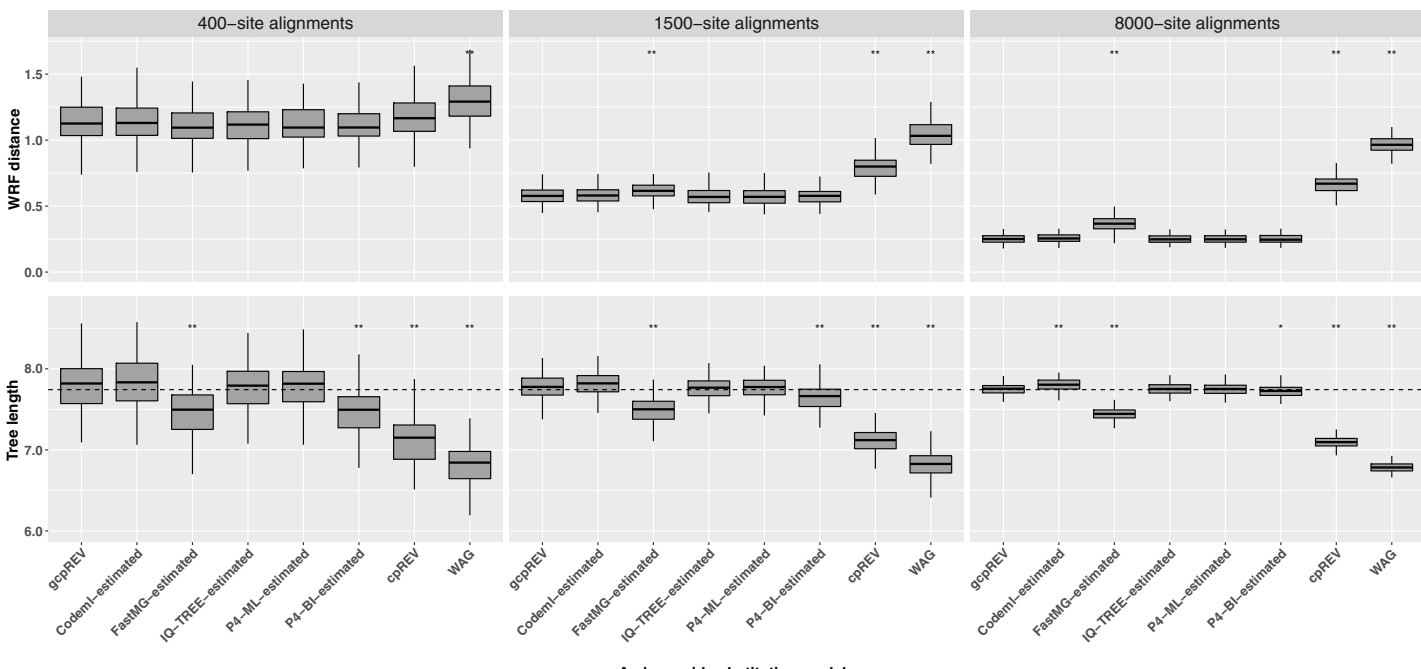

**Figure 2** **Box-plots of weighted Robinson–Foulds (WRF) distances between the simulation tree and the optimal ML trees and the optimal tree lengths.** The ML trees were inferred using data-specific models, the gcpREV (simulation model), cpREV, and WAG (+$\Gamma_4$+$F_{mod}$). The simulation tree length is 7.74 substitutions per site (dashed line). The statistical test evaluates the null hypothesis of no difference between the means of the tree metrics of the optimal trees derived using the simulation model and the means of tree metrics derived from the estimated models, cpREV, and WAG. The null hypothesis was rejected with a P-value (P) significant at <0.05 (*) and <0.01 (**), under a two-tailed t-test. The WRF distances and tree length differences computed from the 1500-site alignments using the FastMG-estimated and P4-BI-estimated models, respectively, were assessed using the Wilcox test because the assumption of normality was rejected according to the Shapiro–Wilk test.

using the Codeml-, IQ-TREE-, and P4-estimated (ML and BI) models converged with the simulation tree length, as the alignment length increased. The optimal trees resulting from IQ-TREE- and P4-ML-estimated models were the closest to the simulation tree and gcpREV derived trees (P > 0.05) about the mean tree length. The Codeml-estimated models also inferred tree lengths close to the simulation length and not significantly different from the gcpREV derived trees, except when computed using the 8,000-site alignments. The optimal trees inferred using the P4-BI estimated models were shorter than the simulation tree and significantly different from the gcpREV derived trees, regardless the alignments length. Nevertheless, their lengths converged strongly with the simulation tree and simulation model trees, as the alignment length increased. The optimal tree lengths inferred from the 1,500-site alignments using the later models were the only values to not have a normal distribution. The optimal ML trees inferred using the FastMG-estimated models were the shortest trees (within data-specific model analyses), significantly different from the gcpREV derived trees, and did not improve with longer alignments.

The estimation of amino-acid substitution models using P4-BI took by far the longest time, taking approximately 34, 68, and 322 h, when inferred from 400, 1,500, and 8,000-site alignments respectively. By contrast, P4-ML (32–756 min), Codeml (14–265 min),

**Table 2 Weighted Robinson-Foulds (WRF) distances between the simulation tree and the optimal ML trees and the optimal tree lengths.**

| Data sets | Amino-acid substitution models | Mean WRF | Δ gcpREV mean WRF | | Mean optimal tree length | Δ simulation tree length | Δ gcpREV optimal mean length | |
|---|---|---|---|---|---|---|---|---|
| | | Subs/site | Subs/site | P | Subs/site | Subs/site | Subs/site | P |
| 400 sites | gcpREV | 1.135 | – | – | 7.797 | 0.054 | – | – |
| | cpREV | 1.179 | 0.044 | $5 \times 10^{-02}$ | 7.122 | −0.621 | −0.675 | $3 \times 10^{-36}$ * |
| | WAG | 1.312 | 0.177 | $4 \times 10^{-12}$ * | 6.819 | −0.924 | −0.978 | $5 \times 10^{-57}$ * |
| | Codeml-estimated | 1.140 | 0.005 | $8 \times 10^{-01}$ | 7.828 | 0.085 | 0.031 | $5 \times 10^{-01}$ * |
| | FastMG-estimated | 1.112 | −0.023 | $3 \times 10^{-01}$ | 7.452 | −0.291 | −0.344 | $3 \times 10^{-12}$ * |
| | IQ-TREE-estimated | 1.117 | −0.018 | $4 \times 10^{-01}$ | 7.756 | 0.013 | −0.040 | $4 \times 10^{-01}$ |
| | P4-ML-estimated | 1.116 | −0.019 | $4 \times 10^{-01}$ | 7.790 | 0.047 | −0.007 | $7 \times 10^{-01}$ |
| | P4-BI-estimated | 1.114 | −0.021 | $3 \times 10^{-01}$ | 7.455 | −0.288 | −0.342 | $2 \times 10^{-13}$ |
| 1,500 sites | gcpREV | 0.582 | – | – | 7.780 | 0.037 | – | – |
| | cpREV | 0.799 | 0.217 | $8 \times 10^{-42}$ * | 7.116 | −0.627 | −0.664 | $5 \times 10^{-75}$ * |
| | WAG | 1.043 | 0.461 | $1 \times 10^{-72}$ * | 6.797 | −0.946 | −0.983 | $9 \times 10^{-99}$ * |
| | Codeml-estimated | 0.583 | 0.001 | $9 \times 10^{-01}$ | 7.811 | 0.068 | 0.032 | $2 \times 10^{-01}$ |
| | FastMG-estimated | 0.621 | 0.039 | $1 \times 10^{-04}$ *a | 7.470 | −0.273 | −0.310 | $3 \times 10^{-25}$ * |
| | IQ-TREE-estimated | 0.574 | −0.008 | $4 \times 10^{-01}$ | 7.754 | 0.011 | −0.026 | $2 \times 10^{-01}$ |
| | P4-ML-estimated | 0.573 | −0.009 | $4 \times 10^{-01}$ | 7.769 | 0.026 | −0.011 | $4 \times 10^{-01}$ |
| | P4-BI-estimated | 0.577 | −0.005 | $6 \times 10^{-01}$ | 7.667 | −0.076 | −0.113 | $3 \times 10^{-08}$ *a |
| 8,000 sites | gcpREV | 0.251 | – | – | 7.750 | 0.007 | – | – |
| | cpREV | 0.666 | 0.416 | $1 \times 10^{-102}$ * | 7.094 | −0.649 | −0.656 | $9 \times 10^{-135}$ * |
| | WAG | 0.968 | 0.717 | $3 \times 10^{-144}$ * | 6.783 | −0.960 | −0.968 | $2 \times 10^{-167}$ * |
| | Codeml-estimated | 0.258 | 0.007 | $1 \times 10^{-01}$ | 7.803 | 0.060 | 0.052 | $2 \times 10^{-06}$ * |
| | FastMG-estimated | 0.368 | 0.118 | $2 \times 10^{-40}$ * | 7.445 | −0.298 | −0.306 | $2 \times 10^{-73}$ * |
| | IQ-TREE-estimated | 0.249 | −0.002 | $8 \times 10^{-01}$ | 7.753 | 0.010 | 0.002 | $8 \times 10^{-01}$ |
| | P4-ML-estimated | 0.249 | −0.001 | $8 \times 10^{-01}$ | 7.749 | 0.006 | −0.001 | $9 \times 10^{-01}$ |
| | P4-BI-estimated | 0.249 | −0.001 | $8 \times 10^{-01}$ | 7.725 | −0.018 | −0.025 | $1 \times 10^{-02}$ * |

**Note:**
The ML trees were inferred using data-specific models, the gcpREV (simulation model), cpREV, and WAG ($+\Gamma_4+F_{mod}$). The null hypotheses of no difference between the means of the tree metrics of the gcpREV derived trees and the means of tree metrics derived from the estimated models, cpREV, and WAG were rejected with a *P*-value (*P*) significant at <0.05 (*), under a two-tailed t-test or under the Wilcox test (a) where the assumption of normality was rejected according to the Shapiro–Wilk test (subs, substitutions).

IQ-TREE (17–155 min), and FastMG (1–3 min), too considerably less time to compute data-specific models (Table S3).

## Re-analysis of published studies using data-specific models

The effectiveness of using data-specific models, estimated using IQ-TREE, was also assessed by reanalysing published empirical data sets and comparing them to commonly-used substitution models (Table 3). The re-analyses of the *Timme, Bachvaroff & Delwiche (2012)* and the *Zeng et al. (2014)* data sets recovered the same topologies as the original published trees, though 2% and 13% longer, respectively, and with likelihood scores of −720,951 and −758,752 log likelihood units, improving on the published scores by 1,486 and 5,972 units, respectively. The optimal ML tree inferred from the *Leliaert et al.*
**Table 3 ML score and length of the optimal trees inferred using data-specific models.**

| Study | Model | Likelihood tree score (log likelihood units) | Tree score improvement (log likelihood units) | Tree length (substitutions per site) | Tree length variance (percentage of substitutions per site) | nRF |
|---|---|---|---|---|---|---|
| *Timme, Bachvaroff & Delwiche (2012)* | Data-specific model $+\Gamma_4$ | −720,951 | 1,486 | 4.7 | 2.0 | 0 |
| *Zeng et al. (2014)* | Data-specific model $+\Gamma_4+$I | −758,752 | 5,972 | 9.4 | 12.8 | 0 |
| *Leliaert et al. (2016)* | Data-specific model $+\Gamma_4$ | −483,209 | 3,590 | 13.9 | 3.2 | 0.05 |
| *Feuda et al. (2017)* | Branch-linked model + data-specific model | −2,305,041 | 16,438 | 19.2 | 15.9 | 0 |
| *Munro et al. (2018)* | Data-specific model $+R_7+$F | −8,070,115 | 43,580 | 9.2 | 10.1 | 0.03 |
| *Schulz et al. (2018)* | Data-specific model $+R_6$ | −470,898 | 1,407 | 76.3 | −10.2 | 0.02 |
| *Schwentner et al. (2017)* (Matrix 1) | Data-specific model $+R_4$ | −443,135 | 1,375 | 6.3 | 1.1 | 0.05 |
| *Toussaint et al. (2018)* (DT369) | Data-specific model $+R_4$ | −620,324 | 10,385 | 1.9 | 6.9 | 0.01 |
| | Branch-unlinked 366-partition + data-specific model | −615,428 | 9,259 | 2.5 | 3.9 | 0.02 |
| | Branch-unlinked 27-partition + data-specific model | −616,082 | 11,171 | 2.0 | 8.5 | 0.01 |
| | Branch-unlinked 27-partition + data-specific models/ partition | −610,590 | 16,664 | 2.0 | 8.9 | 0.01 |
| *Irisarri et al. (2020)* | Data-specific model $+\Gamma_4+$I | −114,615 | 540 | 20.8 | 6.6 | 0 |
| *Koenen et al. (2020)* | Data-specific model $+R_4$ | −571,563 | 38,056 | 7.2 | 4.2 | 0.07 |

**Note:**
Data-specific models were estimated using a GTR ($+\Gamma_{4+}$F$_{est}$) model in IQ-TREE. Normalised Robinson–Foulds (nRF), tree score and length differences were calculated using the optimal trees derived from the data-specific models and the trees derived from the original models.

*(2016)* dataset using a data-specific model had a score of −483,209 log likelihood units, 3,590 units higher than the original model derived tree score, and was 3% longer with a topological distance of 0.05 (nRF) to the tree inferred using the original model.

The re-analysis of the *Chang et al. (2015)* data set using a branch-linked partition model and a data-specific model recovered a tree of score −2,305,041 log likelihood units. *Feuda et al. (2017)* also reanalysed the same data set with a branch-linked partition model; however the published tree score was 16,788 units lower. The topology inferred using the data-specific model was the same as the published tree topology, but 16% longer. The ML tree inferred from the *Munro et al. (2018)* data set using a data-specific model was longer 10% than the original published tree and had a score of −8,070,115 log likelihood units, an improvement of 43,580 units over the published tree score. The resulting tree topology was congruent with the original tree, except for the placement of *Erenna richardi* (nRF = 0.03). The optimal tree resulting from the *Schulz et al. (2018)* data set using a data-specific model had a score of −470,898 log likelihood units, 1,407 units higher than the original published analysis. The tree was 10% shorter than the published tree and had a nRF distance of 0.02.

The re-analysis of the *Schwentner et al. (2017)* recovered an optimal ML tree 1% longer and with a topological distance of 0.05 (nRF) to the original published tree, and with a score of −443,135 log likelihood units, 1,375 units higher than the original tree score.

The re-analyses of the Toussaint (2018) data set included a data-specific model (estimated from the concatenated data set) and different partition schemes, namely, a single partition, a by-locus partition scheme (366 partitions), and a PartitionFinder scheme (27 partitions; from the original study). The reanalysis of the concatenated data set (single partition) had the lowest score (−620,234 log likelihood units) of the data-specific model analyses. Nevertheless, it was higher than any ML tree score of the original analyses. The Bayesian information criteria of the partition analyses using data-specific models were also higher than the original analyses using the same scheme. Additionally, an optimal tree was inferred using the Partition-Finder scheme with separate data-specific models for each partition which had the highest fit to the data, −610,590 log likelihood units.

The data-specific model derived trees had longer lengths, between 4% and 9%, and a nRF varying between 0.007 and 0.022 to the originally published trees. The re-analyses of *Irisarri et al. (2020)* data set recovered a ML tree with a score of −114,615 log likelihood units, fitting the data better than the original model by 541 units. The resulting topology was to the same as the published tree, although 6.6% longer. The ML tree inferred from the *Koenen et al. (2020)* data set was 4.2% longer than the original published tree and had a topological distance of 0.07 (nRF). The tree score was −571,563 log likelihood units, an improvement of 38,056 units over the published tree score.

## DISCUSSION

In this study data-specific substitution models were calculated from simulated sequence data (simulated using a specified substitution model and tree) using five different estimation methods. An accurate model estimation method would be expected to estimate models that are similar to the model used to simulate the data, and result in phylogenetic trees that were similar to the simulation tree with respect to likelihood and topology. Our results showed that data-specific substitution models consistently out-performed empirical, commonly-used, pre-computed substitution matrices. We simulated data using the gcpREV model (*Cox & Foster, 2013*) which was computed for green plant chloroplast data and was intended to more accurately reflect the amino-acid substitution patterns found in green plant chloroplasts when compared to the commonly used cpREV substitution model (*Adachi et al., 2000*) that was estimated from green and non-green plant chloroplasts. The gcpREV model is not commonly-used, and is not used in any of the popular model selection software used to select best-fitting amino-acid substitution models (*e.g.*, ProtTest; *Abascal, Zardoya & Posada, 2005*). The gcpREV simulation model can therefore been seen as a single point in amino-acid substitution space; but it is not a random "distance" from any of the empirical models, just an arbitrary point. Had a simulation model more similar to one of the commonly-used empirical models been chosen, however, the results of the estimated models may not have been distinguishable from those of the empirical model. Nevertheless, given that all of the analyses of published studies (discussed below) showed that data-specific substitution models result in
better-fitting models, we suspect that unless the data are "very close" to being modeled accurately by one of the commonly-used empirical models a data-specific model will be a significantly better fit. Putting aside a caveat regarding data size, given the efficiency at which data-specific exchange rate models can be generated, and their accuracy (see below), it seems unlikely that using an empirical exchange rate model can be justified no matter how closely the data fit the model. The cpREV model was found to be better-fitting than the WAG model to all simulated data, as would be expected. However, the two empirical models, cpREV and WAG, were of much poorer fit to the data than the estimated data-specific substitution models. Indeed, compared to data-specific models, they inferred shorter trees with greater mean WRF and tree length differences to the simulation tree and simulation model (gcpREV) derived trees and failed more often to recover the correct (simulation) topology.

The ML scores of the simulated data on the simulation tree using the data-specific models estimated by each of the five methods were overall similar to each other (Table 1; Fig. S2), and were not statistically different from the ML tree scores of the simulated data using the simulation model. Nevertheless, the tree statistics of the optimal trees sometimes differed between the analyses using the data-specific substitution matrices estimated by the different methods and the those estimated by the simulation model (Table 2; Fig. 2). Specifically, mean WRF distances of optimal trees under the FastMG-estimated model analyses using 1,500- and 8,000-site alignments were significantly different than the WRF values of the optimal trees derived when using the simulation model. Moreover, FastMG-estimated trees were significantly shorter than the optimal tree lengths derived from the simulation model at all data lengths. P4-BI-estimated models also inferred optimal trees that were significantly shorter than the optimal trees derived from the simulation model. By contrast, the remaining estimation methods (Codeml, IQ-TREE, and P4-ML) all estimated optimal trees with mean WRF and lengths statistically indistinguishable from the simulation model derived trees, except for optimal trees estimated by Codeml for 8,000 length data sets which were significantly longer than those of estimated from the simulation model. Codeml uses empirical composition frequencies instead of optimised values, which may explain the lower accuracy compared to the IQ-TREE and P4-ML methods using the longest alignments where more data enable a more precise estimation of the composition.

Data-specific models inferred optimal trees with higher ML scores compared to trees from the simulation model (with the exception of trees inferred from the 8,000-site alignments using FastMG-estimated models, Table 1). This is expected because the data-specific models had free and optimized parameters while the simulation model analysis did not. However, the increase in ML values was not significant for any of the estimation methods, suggesting that even the shortest dataset that we used was sufficient, by this measure, to approximate the simulation model. The difference in ML values between the data-specific models and the simulation model did not show a noticeable trend over dataset sizes (Table 1). For example, the average differences between the P4-ML-estimated trees and the simulation model trees were more or less constant at 91, 100, and 102 log units over increasing dataset sizes, and as mentioned above none of these

differences were significant (with corresponding t-test $P$-values 0.09, 0.32, and 0.59). Increasing $P$-values of this difference over dataset sizes is seen in all estimation methods, and is expected because more data will make the optimized parameters in the data-specific models closer to the simulation model parameters. On the other hand, the alignment length had an impact on the model estimation accuracy of all methods: a putative sample size effect, which is most pronounced in the 400-site alignment analyses, is diminished by more data. In the PCA ordination, the data-specific models estimated from the 400-site alignments clustered further from the simulation model than the models estimated from longer alignments. Moreover, around 20% of optimal trees resulting from the 400-site alignments had a different topology to the simulation tree, while analyses of longer alignment lengths always recovered the simulation topology and had more similar tree and branch lengths of the simulation tree. Despite the observation that 400 sites are still sufficient to estimate accurate data-specific models when considering the lack of significance difference between the ML means compared to those of the simulation model (Table 1), data this length will likely suffer from a higher sample size effect and produce a poorer approximation of the simulation model than those estimated with more data, as suggested by the inaccurate topologies estimated with data this length.

Although the Bayesian model estimation method was relatively accurate compared to ML estimation methods, it took considerable time to implement and would seem impracticable for typical experimental conditions. By contrast, analyses using all four ML estimation methods were completed in reasonable times even for 8,000-site alignments; where we consider reasonable as being assessed in the context of the time needed to generate the data sets and perform typical phylogenetic analyses. However, FastMG, and Codeml to a lesser extent, may be less effective estimation methods where tree and branch lengths are especially important (*e.g.*, dating analyses or molecular rate estimation) for the reasons outlined above. Overall, the IQ-TREE and P4-ML estimation methods appear accurate and time efficient (especially IQ-TREE), and can be recommended for practical use. Indeed, data-specific model estimation appears to be effective even at small sequence data lengths (*e.g.*, 400 sites), as while the estimated models may be inaccurate to some extent perhaps even leading to topological errors, they are also certainly going to be more accurate than pre-computed empirical models unless the data are perchance accurately modeled by one of the very few published models. This interpretation of the simulation results seems to be borne-out by the re-analysis of data from published studies.

The data sets from the published studies we re-analysed varied in numbers of taxa from 14 to 138, had between 5,095 to 365,699 amino acid sites, and included data from nuclear, chloroplast, mitochondria, and viral genomes. In all cases the data-specific models generated in this study were a better fit to the data than the models used in the original studies. Nine of 13 optimal trees inferred using data-specific models were topologically different to the trees published in the original studies (nRF = 0.01–0.07). In addition, optimal trees resulting from data-specific models were longer than the original published trees by between 1.1% and 15.9%, suggesting the that the increased lengths observed in these re-analyses are not artefactual but due to better-fitting models. By contrast, the single viral data set (*Schulz et al., 2018*) had an exceptionally high estimated number of

substitutions per site (90.3) where the data-specific model estimated 10.2% fewer substitutions per site. It has previously been suggested that in most cases commonly used models were inadequate and had a lower ability to describe Flavivirus data sets (*Duchêne, Di Giallonardo & Holmes, 2015*), and indeed the results presented here may suggest that the use of data-specific models may improve accuracy for virus data.

Studies using compound, multi-partition models, such as partition models (*Feuda et al., 2017*; *Toussaint et al., 2018*) and LG4X models (*Schwentner et al., 2017*; *Koenen et al., 2020*), published trees that have lower ML scores than the optimal trees recovered here with data-specific models and a single data partition. The analysis of the chloroplast data set of *Koenen et al. (2020)* using a data-specific model resulted in an optimal tree with a substantial ML score improvement (given the data set length) over the published tree using the LG4X model. This result is likely because the LG4X model (*Le, Dang & Gascuel, 2012*) was generated from nuclear data, and therefore a very poor fit to the chloroplast data despite the seeming sophistication of the model structure. These results indicate that assigning best-fitting empirical amino-acid substitution models according to a partition scheme, or according to sites assigned under a distribution-free scheme (LG4X), may not be sufficient to compensate for the use of inadequately fitting (though best-fitting) empirical substitution models.

Phylogenetic relationships of the optimal trees inferred using data-specific models were mostly congruent with the original studies. However, re-analysis of the *Munro et al. (2018)* data set resulted in an optimal tree where the taxon *Erenna richardi* was placed differently from where it was resolved in the published ML tree. In the original study, its relationship was already noted to be incongruent between ML analyses using a JJT model and BI analyses using the CAT-Poisson model (*Munro et al., 2018*). When using a data-specific model the optimal tree is congruent with the tree from the original BI CAT-Poisson model analyses, which suggests that the amino-acid substitution process of the data cannot not be correctly accommodated by the JTT model in the original ML analysis. In this study we show that data-specific models which have better fit to the data (in terms of likelihood) infer more accurate trees (with respect to topology and branch lengths) than the empirical models. Superficially, this result seems to contrast with recent analyses by *Spielman (2020)* who was unable to distinguish topological accuracy among analyses conducted of simulated amino-acid data with empirical models selected through relative model fit methods. However, as Spielman cautions, the results may be due to the possibility of all models have a poor absolute fit to the data, and indeed, our results suggest a possible reason why that is the case. In our study, we show that the 400 amino-acid sites simulations with a 63.2 mean number of expected substitutions per branch only recovered the correct topology (simulation tree) 78% of the time using the actual simulation model (Table S2). The simulation trees used by *Spielman (2020)* were (with one exception) much larger (60, 305, 274, 200, 179, 103, 70, and 23 taxa) than used on our study (26), while the amino-acid data set lengths were relatively small (262, 497, 564, and 661 sites). More importantly, in terms of diversity, at the longest simulation sequence length of the control simulations (661 sites) only three (Spiralia 81.4; Opisthokonta 100.9; Yeast 145.2—see Supplemental Information) of the eight sets had a mean number of expected substitutions per branch

greater than our 400 site data set (63.2). None of Spielman's simulated control data sets had a mean expected number of substitutions per branch greater than our 1,500 site data set (all were considerably lower). Moreover, Spielman notes that the only simulations to reconstruct the correct topology (simulation tree) were from analyses of simulated data from the Spiralia, Opisthokonta, or Yeast phylogenies, suggesting data size and site diversity may be crucial. While these are broad summary statistics, it does point to the probability that the simulated data sets of Spielman were insufficiently variable and/or too short to distinguish among the models being compared (see also *Del Amparo & Arenas (2023)* for a similar critique).

## CONCLUSIONS

Of the five software methods for computing data-specific protein models we compared, the IQ-TREE and P4 (ML) methods were shown to be the most accurate, with IQ-TREE being the most time efficient of the two. Given the availability of time efficient software to calculate sufficiently accurate data-specific amino-acid substitution models there seems no longer any justification for using pre-computed empirical models in phylogenetic analyses, even when the data are limited (~400-sites). Indeed, the time needed to choose a model from among an assortment of empirical models may be longer than the time needed to compute a data-specific model. Moreover, one could imagine a tool that calculates whether the best-fitting empirical model is a sufficiently good fit to the data when compared to a data-specific model by simulating data and performing a statistical test, but if our results generalise to most data, as we suspect they do, then such analyses are superfluous: just use the already calculated a data-specific substitution model.

### Funding

This study received Portuguese national funds from FCT—Foundation for Science and Technology—through project UIDB/04326/2020, UIDP/04326/2020 and LA/P/0101/2020, and from the operational programmes CRESC Algarve 2020 and COMPETE 2020 through projects EMBRC.PT ALG-01-0145-FEDER-022121 and BIODATA.PT ALG-01-0145-FEDER-022231. FCT also provided funding through the individual research grant (SFRH/BD/134422/2017) to João M. Brazão. The funders had no role in study design, data collection and analysis, decision to publish, or preparation of the manuscript.

### Grant Disclosures

The following grant information was disclosed by the authors:
FCT—Foundation for Science and Technology: UIDB/04326/2020, UIDP/04326/2020 and LA/P/0101/2020.
CRESC Algarve 2020 and COMPETE 2020: EMBRC.PT ALG-01-0145-FEDER-022121 and BIODATA.PT ALG-01-0145-FEDER-022231. FCT: SFRH/BD/134422/2017.

## Competing Interests

The authors declare that they have no competing interests.

## Author Contributions

- João M. Brazão performed the experiments, analyzed the data, prepared figures and/or tables, authored or reviewed drafts of the article, and approved the final draft.
- Peter G. Foster conceived and designed the experiments, authored or reviewed drafts of the article, and approved the final draft.
- Cymon J. Cox conceived and designed the experiments, authored or reviewed drafts of the article, and approved the final draft.

## Data Availability

The data products (protein alignments, calculated substitution models, and trees) are available from Zenodo: João Brazão. (2023). Data-specific substitution models improve protein-based phylogenetics—data [Data set]. Zenodo. https://doi.org/10.5281/zenodo.7628408.

The novel scripts used to make calculations and a machine actionable RO-Crate metadata specification are available from GitHub: https://github.com/joaobrazao/Data-specific-substitution-models-improve-protein-based-phylogenetics.

## Supplemental Information

Supplemental information for this article can be found online at http://dx.doi.org/10.7717/peerj.15716#supplemental-information.

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
