# Peer review of "Data-specific substitution models improve protein-based phylogenetics"

_PeerJ, doi:10.7717/peerj.15716_

## Round 0.1 · original submission · Major Revisions

Dear Dr. Brazão and colleagues:

Thanks for submitting your manuscript to PeerJ. I have now received three independent reviews of your work, and as you will see, the reviewers raised some concerns about the research. Despite this, these reviewers are optimistic about your work and the potential impact it will have on research studying methods in phylogeny estimation. Thus, I encourage you to revise your manuscript, accordingly, taking into account all of the concerns raised by the reviewers.

Your manuscript needs to be restructured and streamlined to improve clarity and presentation. The reviewers collectively provide many helpful tips. Also, reviewer 1 brings up some concerns about aspects of your modelling, especially regarding rate heterogeneity and gap treatment. Please address these issues and also ensure your workflow is repeatable and all scripts are made public.

Please note that reviewer 3 has included a marked-up version of your manuscript.

There are many suggestions, which I am sure will greatly improve your manuscript once addressed.

I look forward to seeing your revision, and thanks again for submitting your work to PeerJ.

Good luck with your revision,

Best,

-joe

·

Basic reporting

The study “Data-specific substitution models improve protein-based phylogenetics” by Brazao, Foster and Cox provides an evaluation of several methods to estimate substitution models (the exchangeability matrix and amino acid frequencies at the equilibrium when available). I think that this study is a good contribution to the field and it can be useful for readers interested in estimating substitution models for particular data. I have several comments that I think should be considered to improve the study. I recommend revisions.

Major comments

The conclusions of the abstract and conclusions section do not clearly indicate what is the most accurate method, among the evaluated methods, to estimate a substitution model of protein evolution. After reading the manuscript, there is not a clear answer to the main question of this study: what is the most accurate method to estimate a substitution model of protein evolution (whatever is the time required to estimate it)?. If a reader wants to use the most accurate program (among those evaluated in this study), what is that one?. I think this clarification in the abstract in the conclusions section is fundamental. Otherwise the study does not provide a really advance.

The manuscript indicates “Exact commands and parameter values for each analysis are provided in the Supplementary Information”, however I could not find this in the supplementary files. In which supplementary file is this information?

The study is based on simulations done with the model gcpREV, which also is not commonly used in practice. What about if simulations are done with a model different from gcpREV?, could change that model in the simulations affect the results and conclusions?

Results shown in Figure 2 show that most of the models overlap in WRF distance and tree length, only the models cpREV and WAG slightly differ from the others in data with many sites. Does it mean that the method used to estimate the substitution model does matter because most of the models produce similar results (WRF distance and tree length)?

The comparison among models (i.e., Table 1) should consider not only the likelihoods but also the number of parameters of the model (as done with AIC and BIC). Thus I could not understand why statistical tests (i.e., mentioned in the caption of Table 1) are based on the likelihoods, I think instead they should use AIC and/or BIC scores.
Also I think Tables 1, 3 and S1 should include AIC and/or BIC scores.

The manuscript indicates in several places that “Data-specific models performed better than the empirical models”. For example it is mentioned in the lines 93-94 of the introduction. There are two points that I would like to mention about this.
First, I agree that data-specific models can perform better than empirical models, this is for example supported by the study, https://www.mdpi.com/2073-4425/13/1/61
This is because a model based on data similar to the query data better fits with the query data than a model based on data different to the query data, but supporting that with examples I think is more appropriate.
In any case and very important, it should be clear that data-specific models should be based on data that is not used for training the model, otherwise the results derived could be biased.
Second, I think it should be clear that general models can still be useful (i.e., in contrast to the line 352-353: it seems unlikely that using an empirical model can be justified no matter how well the data are modeled by a published empirical model). General models can be useful for certain analyses at a higher scale where large diversity is fundamental, such as the reconstruction of the tree of life, see https://www.mdpi.com/2079-7737/12/2/282
I think these aspects should be discussed in the manuscript from a more general biological perspective, citing the mentioned studies and not only focused on the findings of this study that are limited by its studied data.

What is the influence of the protein diversity in the data used to estimate the substitution model on the error on the estimated substitution model? A higher diversity in the training data can produce more error in the estimated substitution model?. I think this aspect can be interesting for readers.

The introduction indicates that a poor fit of the model to the data will affect the accuracy of tree reconstruction. However, a recent paper indicated the opposite, https://academic.oup.com/mbe/article/37/7/2110/5810088
In my opinion, substitution model selection matters for many phylogenetic analyses such as ancestral sequence reconstruction (among other analyses, including phylogenetic tree reconstruction),
https://academic.oup.com/mbe/article/39/7/msac144/6628884
I would like to know the opinion of the authors about these aspects. Also I think these aspects should be mentioned in the manuscript (including the mentioned papers) because if one aims to estimate a substitution model I guess assumes that estimate it will be positive (not just neutral) for the accuracy of the phylogenetic analyses derived.

Minor comments

I could not understand the meaning of some points present in Figure 1. For example, what is the difference between “Codeml_8000” and “Codeml_1500”?. This should be clarified in the caption.

I think the files with tables or figures of the supplementary material should include a caption presenting what is shown in the corresponding table or figure. I could not download the captions from the journal, only the files.

Experimental design

All my comments are presented in "1. Basic reporting".

Validity of the findings

All my comments are presented in "1. Basic reporting".

Additional comments

All my comments are presented in "1. Basic reporting".

Reviewer 2 ·

Basic reporting

The authors systemically evaluated the amino acid substitution models. The study compared the performance of five data-specific methods: Codeml, FastMG, IQ-TREE, P4 (maximum likelihood), and P4 (Bayesian inference), as well as two commonly-used empirical models, cpREV and WAG. The manuscript is well-written and easy to follow, but there are a few points that need to be improved.

Experimental design

The methods used in the study are described in sufficient detail. But there is no source code available for replication purposes. Can you submit the analysis code to a code repository such as GitHub? so other researchers of interest can evaluate or develop new data-specific models?

Validity of the findings

After comprehensive evaluations, the authors provided valuable suggestions on amino acid substitution model choices.

Additional comments

1) The font size for Figure 1 is way too small, please enlarge it.
2) For Figure 2, I suggest moving group FastMG-estimated from the third position to the sixth position. Or authors may plot the x-axis labels using ranking by Tree length.
3) on line 111, what does clustering distance (PCA) mean?

Reviewer 3 ·

Basic reporting

Please refer to the attached pdf file.

Experimental design

Please refer to the attached pdf file.

Validity of the findings

Please refer to the attached pdf file.

Additional comments

Please refer to the attached pdf file.

Annotated reviews are not available for download in order to protect the identity of reviewers who chose to remain anonymous.

---

## Round 0.2 · Minor Revisions

Dear Dr. Brazão and colleagues:

Thanks for revising your manuscript. The reviewers are very satisfied with your revision (as am I). Great! However, there are a couple of suggestions to entertain. Please address these ASAP so we may move towards acceptance of your work.

Best,

-joe

·

Basic reporting

The new version of the study “Data-specific substitution models improve protein-based phylogenetics” by Brazao, Foster, and Cox is much more clear and detailed. I only have two minor comments to the authors as optional, in my view not mandatory to accept this study for publication.

Minor comments

In the previous revision, I indicated “The study is based on simulations done with the model gcpREV, which also is not commonly used in practice. What about if simulations are done with a model different from gcpREV? could change that model in the simulations affect the results and conclusions?”
The authors answered that they believe that changing the model used for the simulations would not lead to substantively different conclusions, but it was not tested. I agree with the authors but I was wondering about adding a short test to confirm it. It could be based on simulations with a commonly used model such as JTT, WAG or LG. Just in case the authors find it convenient.

Another limitation is not exploring the influence of genetic diversity. The simulations were only based on a single tree. I think the effect of genetic diversity could be easily investigated for example by multiplying all the branch lengths by a factor, for example 0.5 and 2 to study two additional levels of genetic diversity. Again just in case the authors find it convenient.

Experimental design

All my comments are presented in "1. Basic reporting".

Validity of the findings

All my comments are presented in "1. Basic reporting".

Additional comments

All my comments are presented in "1. Basic reporting".

Reviewer 2 ·

Basic reporting

The authors have thoroughly incorporated my feedback from the previous round of review to the best of their abilities. I have no further suggestions to make at this time.

Experimental design

no comment

Validity of the findings

no comment

Reviewer 3 ·

Basic reporting

My comments were all addressed well. I am happy with their revised manuscript.

Experimental design

My comments were all addressed well. I am happy with their revised manuscript.

Validity of the findings

My comments were all addressed well. I am happy with their revised manuscript.

Additional comments

My comments were all addressed well. I am happy with their revised manuscript.

---

## Round 0.3 · accepted · Accept

Dear Dr. Brazão and colleagues:

Thanks for revising your manuscript based on the concerns raised by the reviewers. I now believe that your manuscript is suitable for publication. Congratulations! I look forward to seeing this work in print, and I anticipate it being an important resource for groups studying methods in phylogeny estimation. Thanks again for choosing PeerJ to publish such important work.

Best,

-joe